# Adversarial Examples that Fool both Computer Vision and Time-Limited Humans

**Gamaleldin F. Elsayed**[*]
Google Brain
gamaleldin.elsayed@gmail.com

**Shreya Shankar**
Stanford University

**Brian Cheung**
UC Berkeley

**Nicolas Papernot**
Pennsylvania State University

**Alexey Kurakin**
Google Brain

**Ian Goodfellow**
Google Brain

**Jascha Sohl-Dickstein**
Google Brain
jaschasd@google.com

## Abstract

Machine learning models are vulnerable to **adversarial examples**: small changes to images can cause computer vision models to make mistakes such as identifying a school bus as an ostrich. However, it is still an open question whether humans are prone to similar mistakes. Here, we address this question by leveraging recent techniques that transfer adversarial examples from computer vision models with known parameters and architecture to other models with unknown parameters and architecture, and by matching the initial processing of the human visual system. We find that adversarial examples that strongly transfer across computer vision models influence the classifications made by time-limited human observers.

## 1 Introduction

Machine learning models are easily fooled by adversarial examples: inputs optimized by an adversary to produce an incorrect model classification [39, 3]. In computer vision, an adversarial example is usually an image formed by making small perturbations to an example image. Many algorithms for constructing adversarial examples [39, 13, 33, 24, 27] rely on access to both the architecture and the parameters of the model to perform gradient-based optimization on the input. Without similar access to the brain, these methods do not seem applicable to constructing adversarial examples for humans.

One interesting phenomenon is that adversarial examples often transfer from one model to another, making it possible to attack models that an attacker has no access to [39, 26]. This naturally raises the question of whether humans are susceptible to these adversarial examples. Clearly, humans are prone to many cognitive biases and optical illusions [17], but these generally do not resemble small perturbations of natural images, nor are they currently generated by optimization of a ML loss function. Thus the current understanding is that this class of transferable adversarial examples has no effect on human visual perception, yet no thorough empirical investigation has yet been performed.

A rigorous investigation of the above question creates an opportunity both for machine learning to gain knowledge from neuroscience, and for neuroscience to gain knowledge from machine learning. Neuroscience has often provided existence proofs for machine learning—before we had working object recognition algorithms, we hypothesized it should be possible to build them because the human

---

[*]Work done as a member of the Google AI Residency program (g.co/airesidency).

brain can recognize objects. See Hassabis et al. [15] for a review of the influence of neuroscience on artificial intelligence. If we knew conclusively that the human brain could resist a certain class of adversarial examples, this would provide an existence proof for a similar mechanism in machine learning security. If we knew conclusively that the brain can be fooled by adversarial examples, then machine learning security research should perhaps shift its focus from designing models that are robust to adversarial examples [39, 13, 32, 42, 40, 27, 19, 5] to designing systems that are secure despite including non-robust machine learning components. Likewise, if adversarial examples developed for computer vision affect the brain, this phenomenon discovered in the context of machine learning could lead to a better understanding of brain function.

In this work, we construct adversarial examples that transfer from computer vision models to the human visual system. In order to successfully construct these examples and observe their effect, we leverage three key ideas from machine learning, neuroscience, and psychophysics. First, we use the recent **black box** adversarial example construction techniques that create adversarial examples for a target model without access to the model's architecture or parameters. Second, we adapt machine learning models to mimic the initial visual processing of humans, making it more likely that adversarial examples will transfer from the model to a human observer. Third, we evaluate classification decisions of human observers in a time-limited setting, so that even subtle effects on human perception are detectable. By making image presentation sufficiently brief, humans are unable to achieve perfect accuracy even on clean images, and small changes in performance lead to more measurable changes in accuracy. Additionally, a brief image presentation limits the time in which the brain can utilize recurrent and top-down processing pathways [34], and is believed to make the processing in the brain more closely resemble that in a feedforward artificial neural network.

We find that adversarial examples that transfer across computer vision models *do* successfully influence the perception of human observers, thus uncovering a new class of illusions that are shared between computer vision models and the human brain.

## 2 Background and Related Work

### 2.1 Adversarial Examples

Goodfellow et al. [12] define adversarial examples as "inputs to machine learning models that an attacker has intentionally designed to cause the model to make a mistake." In the context of visual object recognition, adversarial examples are images usually formed by applying a small perturbation to a naturally occurring image in a way that breaks the predictions made by a machine learning classifier. See Figure Supp.1a for a canonical example where adding a small perturbation to an image of a panda causes it to be misclassified as a gibbon. This perturbation is small enough to be imperceptible (i.e., it cannot be saved in a standard png file that uses 8 bits because the perturbation is smaller than $1/255$ of the pixel dynamic range). This perturbation relies on carefully chosen structure based on the parameters of the neural network—but when magnified to be perceptible, human observers cannot recognize any meaningful structure. Note that adversarial examples also exist in other domains like malware detection [14], but we focus here on image classification tasks.

Two aspects of the definition of adversarial examples are particularly important for this work:

1. Adversarial examples are designed to cause a *mistake*. They are not (as is commonly misunderstood) defined to differ from human judgment. If adversarial examples were defined by deviation from human output, it would by definition be impossible to make adversarial examples for humans. On some tasks, like predicting whether input numbers are prime, there is a clear objectively correct answer, and we would like the model to get the correct answer, not the answer provided by humans (time-limited humans are probably not very good at guessing whether numbers are prime). It is challenging to define what constitutes a mistake for visual object recognition. After adding a perturbation to an image it likely no longer corresponds to a photograph of a non-contrived physical scene. Furthermore, it is philosophically difficult to define the real object class for an image that is not a picture of a real object. In this work, we assume that an adversarial image is misclassified if the output label differs from the human-provided label of the clean image that was used as the starting point for the adversarial image. We make small adversarial perturbations and we assume that these small perturbations are insufficient to change the true class.

2. Adversarial examples are not (as is commonly misunderstood) defined to be imperceptible. If this were the case, it would be impossible by definition to make adversarial examples for humans, because changing the human's classification would constitute a change in what the human perceives (e.g., see Figure Supp.1b,c).

### 2.1.1 Clues that Transfer to Humans is Possible

Some observations give clues that transfer to humans may be possible. Adversarial examples are known to transfer across machine learning models, which suggest that these adversarial perturbations may carry information about target adversarial classes. Adversarial examples that fool one model often fool another model with a different architecture [39], another model that was trained on a different training set [39], or even trained with a different algorithm [30] (e.g., adversarial examples designed to fool a convolution neural network may also fool a decision tree). The transfer effect makes it possible to perform black box attacks, where adversarial examples fool models that an attacker does not have access to [39, 31]. Kurakin et al. [24] found that adversarial examples transfer from the digital to the physical world, despite many transformations such as lighting and camera effects that modify their appearance when they are photographed in the physical world. Liu et al. [26] showed that the transferability of an adversarial example can be greatly improved by optimizing it to fool *many* machine learning models rather than one model: an adversarial example that fools five models used in the optimization process is more likely to fool an arbitrary sixth model.

Moreover, recent studies on stronger adversarial examples that transfer across multiple settings have sometimes produced adversarial examples that appear more meaningful to human observers. For instance, a cat adversarially perturbed to resemble a computer [2] while transfering across geometric transformations develops features that appear computer-like (Figure Supp.1b), and the 'adversarial toaster' from Brown et al. [4] possesses features that seem toaster-like (Figure Supp.1c). This development of human-meaningful features is consistent with the adversarial example carrying true feature information and thus coming closer to fooling humans, if we acounted for the notable differences between humans visual processing and computer vision models (see section 2.2.2)

## 2.2 Biological and Artificial Vision

### 2.2.1 Similarities

Recent research has found similarities in representation and behavior between deep convolutional neural networks (CNNs) and the primate visual system [6]. This further motivates the possibility that adversarial examples may transfer from computer vision models to humans. Activity in deeper CNN layers has been observed to be predictive of activity recorded in the visual pathway of primates [6, 43]. Reisenhuber and Poggio [36] developed a model of object recognition in cortex that closely resembles many aspects of modern CNNs. Kummerer et al. [21, 22] showed that CNNs are predictive of human gaze fixation. Style transfer [10] demonstrated that intermediate layers of a CNN capture notions of artistic style which are meaningful to humans. Freeman et al. [9] used representations in a CNN-like model to develop psychophysical metamers, which are indistinguishable to humans when viewed briefly and with carefully controlled fixation. Psychophysics experiments have compared the pattern of errors made by humans, to that made by neural network classifiers [11, 35].

### 2.2.2 Notable Differences

Differences between machine and human vision occur early in the visual system. Images are typically presented to CNNs as a static rectangular pixel grid with constant spatial resolution. The primate eye on the other hand has an eccentricity dependent spatial resolution. Resolution is high in the fovea, or central $\sim 5°$ of the visual field, but falls off linearly with increasing eccentricity [41]. A perturbation which requires high acuity in the periphery of an image, as might occur as part of an adversarial example, would be undetectable by the eye, and thus would have no impact on human perception. Further differences include the sensitivity of the eye to temporal as well as spatial features, as well as non-uniform color sensitivity [25]. Modeling the early visual system continues to be an area of active study [29, 28]. As we describe in section 3.1.2, we mitigate some of these differences by using a biologically-inspired image input layer.

Beyond early visual processing, there are more major computational differences between CNNs and the human brain. All the CNNs we consider are fully feedforward architectures, while the

visual cortex has many times more feedback than feedforward connections, as well as extensive recurrent dynamics [29]. Possibly due to these differences in architecture, humans have been found experimentally to make classification mistakes that are qualitatively different than those made by deep networks [8]. Additionally, the brain does not treat a scene as a single static image, but actively explores it with saccades [18]. As is common in psychophysics experiments [20], we mitigate these differences in processing by limiting both the way in which the image is presented, and the time which the subject has to process it, as described in section 3.2.

## 3   Methods

Section 3.1 details our machine learning vision pipeline. Section 3.2 describes our psychophysics experiment to evaluate the impact of adversarial images on human subjects.

### 3.1   The Machine Learning Vision Pipeline

#### 3.1.1   Dataset

In our experiment, we used images from ImageNet [7]. ImageNet contains 1,000 highly specific classes that typical people may not be able to identify, such as "Chesapeake Bay retriever". Thus, we combined some of these fine classes to form six coarse classes we were confident would be familiar to our experiment subjects ({dog, cat, broccoli, cabbage, spider, snake}). We then grouped these six classes into the following groups: (i) **Pets** group (dog and cat images); (ii) **Hazard** group (spider and snake images); (iii) **Vegetables** group (broccoli and cabbage images).

#### 3.1.2   Ensemble of Models

We constructed an ensemble of ten CNN models trained on ImageNet. Each model is an instance of one of these architectures: Inception V3, Inception V4, Inception ResNet V2, ResNet V2 50, ResNet V2 101, and ResNet V2 152 [38, 37, 16]. To better match the initial processing of human visual system, we prepend each model with a retinal layer, which pre-processes the input to incorporate some of the transformations performed by the human eye. In that layer, we perform an eccentricity dependent blurring of the image to approximate the input which is received by the visual cortex of human subjects through their retinal lattice. The details of this retinal layer are described in Appendix B. We use eccentricity-dependent spatial resolution measurements (based on the macaque visual system) [41], along with the known geometry of the viewer and the screen, to determine the degree of spatial blurring at each image location. This limits the CNN to information which is also available to the human visual system. The layer is fully differentiable, allowing gradients to backpropagate through the network when running adversarial attacks. Further details of the models and their classification performance are provided in Appendix E.

#### 3.1.3   Generating Adversarial Images

For a given image group, we wish to generate targeted adversarial examples that strongly transfer across models. This means that for a class pair $(A, B)$ (e.g., $A$: cats and $B$: dogs), we generate adversarial perturbations such that models will classify perturbed images from $A$ as $B$; similarly, we perturbed images from $B$ to be classified as $A$. A different perturbation is constructed for each image; however, the $\ell_\infty$ norm of all perturbations are constrained to be equal to a fixed $\epsilon$.

Formally: given a classifier, which assigns probability $P(y \mid X)$ to each coarse class $y$ given an input image $X$, a specified target class $y_{\text{target}}$ and a perturbation magnitude $\epsilon$, we want to find the image $X_{adv}$ that minimizes $-\log(P(y_{\text{target}} \mid X_{adv}))$ with the constraint that $||X_{adv} - X||_\infty = \epsilon$. See Appendix C for details on computing the coarse class probabilities $P(y \mid X)$. With the classifier's parameters, we can perform iterated gradient descent on $X$ in order to generate our $X_{adv}$ (see Appendix D). This iterative approach is commonly employed to generate adversarial images [24].

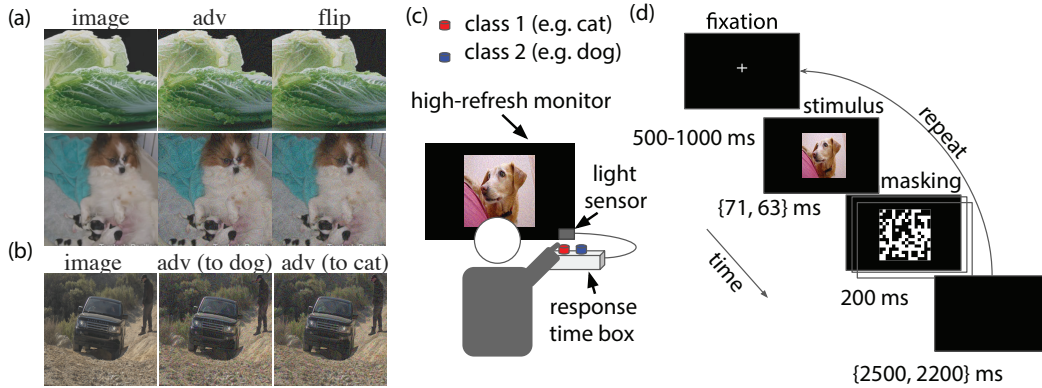

Figure 1: **Experiment setup and task.** (a) examples images from the conditions (`image`, `adv`, and `flip`). Top: `adv` targeting broccoli class. bottom: `adv` targeting cat class. See definition of conditions at Section 3.2.2. (b) example images from the `false` experiment condition. (c) Experiment setup and recording apparatus. (d) Task structure and timings. The subject is asked to repeatedly identify which of two classes (e.g. dog vs. cat) a briefly presented image belongs to. The image is either adversarial, or belongs to one of several control conditions. See Section 3.2 for details.

## 3.2 Human Psychophysics Experiment

38 subjects with normal or corrected vision participated in the experiment. Subjects gave informed consent to participate, and were awarded a reasonable compensation for their time and effort[2].

### 3.2.1 Experimental Setup

Subjects sat on a fixed chair 61 cm away from a high refresh-rate computer screen (ViewSonic XG2530) in a room with dimmed light (Figure 1c). Subjects were asked to classify images that appeared on the screen into one of two classes (two alternative forced choice) by pressing buttons on a response time box (LOBES v5/6:USTC) using two fingers on their right hand. The assignment of classes to buttons was randomized for each experiment session, and labels were placed next to the buttons to prevent confusion. Each trial started with a fixation cross displayed in the middle of the screen for $500 - 1000$ ms. Subjects were instructed to direct their gaze to the center of this cross (Figure 1d). After the fixation period, an image of size $15.24$ cm $\times$ $15.24$ cm ($14.2°$ visual angle) was presented briefly at the center of the screen for a period of 63 ms (71 ms for some sessions). The image was followed by a sequence of ten high contrast binary random masks, each displayed for 20 ms (see example in Figure 1d). Subjects were asked to classify the object in the image (e.g., cat vs. dog) by pressing one of two buttons starting at the image presentation time and lasting until 2200 ms (or 2500 ms for some sessions) after the mask was turned off. The waiting period to start the next trial was of the same duration whether subjects responded quickly or slowly. Realized exposure durations were $\pm 4$ ms from the times reported above, as measured by a photodiode and oscilloscope in a separate test experiment. Each subject's response time was recorded by the response time box relative to the image presentation time (monitored by a photodiode). In the case where a subject pressed more than one button in a trial, only the class corresponding to their first choice was considered. Each subject completed between 140 and 950 trials. Further, subjects performed one or more demo trials at the start of the session, to gain familiarity with the task. During the demo trials only, image presentation time was long, and subjects were given feedback on the correctness of their choice.

### 3.2.2 Experiment Conditions

Each experimental session included only one of the image groups (Pets, Vegetables or Hazard). For each group, images were presented in one of four conditions as follows:

- `image`: images from the ImageNet training set (rescaled to the $[40, 255 - 40]$ range to avoid clipping when adversarial perturbations are added; see Figure 1a left) ).

- `adv`: we added adversarial perturbation $\delta_{adv}$ to `image`, crafted to cause machine learning models to misclassify `adv` as the opposite class in the group (e.g., if `image` was originally a cat, we perturbed the image to be classified as a dog). We used a perturbation size large enough to be noticeable by humans on the computer screen but small with respect to the image intensity scale ($\epsilon = 32$; see Figure 1a middle). In other words, we chose $\epsilon$ to be large (to improve the chances of adversarial examples transfer to time-limited human) but kept it small enough that the perturbations are class-preserving (as judged by a no-limit human).

- `flip`: similar to `adv`, but the adversarial perturbation ($\delta_{adv}$) is flipped vertically before being added to `image`. This is a control condition, chosen to have nearly identical perturbation statistics to the `adv` condition (see Figure 1a right).

- `false`: in this condition, subjects are forced to make a mistake. To show that adversarial perturbations *actually control the chosen class*, we include this condition where neither of the two options available to the subject is correct, so their accuracy is always zero. We test whether adversarial perturbations can influence which of the two wrong choices they make. We show a random image from an ImageNet class other than the two classes in the group, and adversarially perturb it toward one of the two classes in the group. The subject must then choose from these two classes. For example, we might show an airplane adversarially perturbed toward the dog class, while a subject is in a session classifying images as cats or dogs. We used a slightly larger perturbation in this condition ($\epsilon = 40$; see Figure 1b).

The conditions (`image`, `adv`, `flip`) are ensured to have balanced number of trials within a session by either uniformly sampling the condition type in some of the sessions or randomly shuffling a sequence with identical trial counts for each condition in other sessions. The number of trials for each class in the group was also constrained to be equal. Similarly for the `false` condition the number of trials adversarially perturbed towards class 1 and class 2 were balanced for each session. To reduce subjects using strategies based on overall color or brightness distinctions between classes, we pre-filtered the dataset to remove images that showed an obvious effect of this nature. Notably, in the pets group we excluded images that included large green lawns or fields, since in almost all cases these were photographs of dogs. See Appendix F for images used in the experiment for each coarse class. For example images for each condition, see Figures Supp.5 through Supp.8.

## 4 Results

### 4.1 Adversarial Examples Transfer to Computer Vision Models

We first assess the transfer of our constructed images to two test models that were not included in the ensemble used to generate adversarial examples. These test models are an adversarially trained Inception V3 model [23] and a ResNet V2 50 model. Both models perform well ($> 75\%$ accuracy) on clean images. Attacks in the `adv` and `false` conditions succeeded against the test models between $57\%$ and $89\%$ of the time, depending on image class and experimental condition. The `flip` condition changed the test model predictions on fewer than $1.5\%$ of images in all conditions, validating its use as a control. See Tables Supp.3 - Supp.6 for accuracy and attack success measurements on both train and test models for all experimental conditions.

### 4.2 Adversarial Examples Transfer to Humans

We now show that adversarial examples transfer to time-limited humans. One could imagine that adversarial examples merely degrade image quality or discard information, thus increasing error rate. To rule out this possibility, we begin by showing that for a fixed error rate (in a setting where the human is forced to make a mistake), adversarial perturbations influence the human choice among two incorrect classes. Then, we demonstrate that adversarial examples increase the error rate.

#### 4.2.1 Influencing the Choice between two Incorrect Classes

As described in Section 3.2.2, we used the `false` condition to test whether adversarial perturbations can influence which of two incorrect classes a subject chooses (see example images in Figure Supp.5).

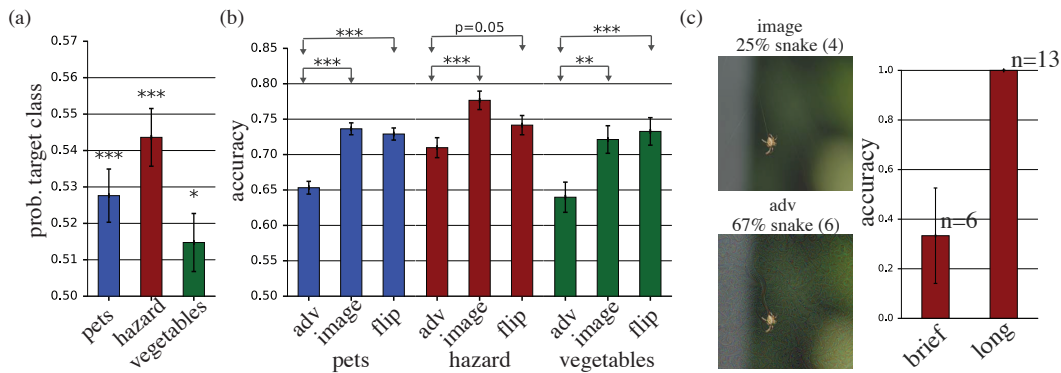

Figure 2: **Adversarial images transfer to humans.** (a) By adding adversarial perturbations to an image, we are able to bias which of two incorrect choices subjects make. Plot shows probability of choosing the adversarially targeted class when the true image class is not one of the choices that subjects can report (`false` condition), estimated by averaging the responses of all subjects (two-tailed t-test relative to chance level $0.5$). (b) Adversarial images cause more mistakes than either clean images or images with the adversarial perturbation flipped vertically before being applied. Plot shows probability of choosing the true image class, when this class is one of the choices that subjects can report, averaged across all subjects. Accuracy is significantly less than 1 even for clean images due to the brief image presentation time. (error bars $\pm$ SE; *: $p < 0.05$; **: $p < 0.01$; ***: $p < 0.001$) (c) A spider image that time-limited humans frequently perceived as a snake (top parentheses: number of subjects tested on this image). right: accuracy on this adversarial image when presented briefly compared to when presented for long time (long presentation is based on a post-experiment survey of 13 participants).

We measured our effectiveness at changing the perception of subjects using the rate at which subjects reported the adversarially targeted class. If the adversarial perturbation were completely ineffective we would expect the choice of targeted class to be uncorrelated with the subject's reported class. The average rate at which the subject chooses the target class metric would be $0.5$ as each `false` image is perturbed to class 1 or class 2 in the group with equal probability. Figure 2a shows the probability of choosing the target class averaged across all subjects for all three experiment groups. In all cases, the probability was significantly above the chance level of $0.5$. This demonstrates that the adversarial perturbations generated using CNNs biased human perception towards the targeted class. This effect was stronger for the the hazard, then pets, then vegetables group. This difference in probability among the class groups was significant ($p < 0.05$; Pearson Chi-2 GLM test).

We also observed a significant difference in the mean response time between the class groups ($p < 0.001$; one-way ANOVA test; see Figure Supp.2a). Interestingly, the response time pattern across image groups (Figure Supp.2a)) was inversely correlated to the perceptual bias pattern (Figure 2a)) (Pearson correlation $= -1$, $p < 0.01$; two-tailed Pearson correlation test). In other words, subjects made quicker decisions for the hazard group, then pets group, and then vegetables group. This is consistent with subjects being more confident in their decision when the adversarial perturbation was more successful in biasing subjects perception. This inverse correlation between attack success and response time was observed within group, as well as between groups (Figure Supp.3).

### 4.2.2 Adversarial Examples Increase Human Error Rate

We demonstrated that we are able to bias human perception to a target class when the true class of the image is not one of the options that subjects can choose. Now we show that adversarial perturbations can be used to cause the subject to choose an incorrect class even though the correct class is an available response. As described in Section 3.2.2, we presented `image`, `flip`, and `adv`.

Most subjects had lower accuracy in `adv` than `image` (Table Supp.1). This is also reflected on the average accuracy across all subjects significantly lower for the `adv` than `image` (Figure 2b).

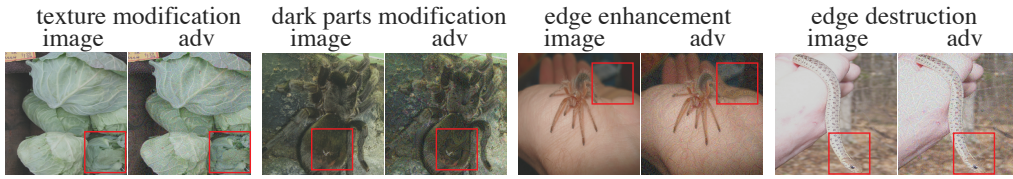

Figure 3: **Examples of the types of manipulations performed by the adversarial attack.** See Figures Supp.6 through Supp.8 for additional examples of adversarial images. Also see Figure Supp.5 for adversarial examples from the `false` condition.

The above result in isolation might be simply explained by the fact that images from the `adv` condition include perturbations whereas images from the `image` condition are unaltered. While this issue is largely addressed by the `false` experiment results in Section 4.2.1, to provide a further control we also evaluated accuracy on `flip` images. This control case uses perturbations with identical statistics to `adv` up to a flip of the vertical axis. However, this control breaks the pixel-to-pixel correspondence between the adversarial perturbation and the image. The majority of subjects had lower accuracy in the `adv` condition than in the `flip` condition (Table Supp.1). When averaging across all trials, this effect was very significant for the pets and vegetables group ($p < 0.001$), and less significant for the hazard group ($p = 0.05$) (Figure 2b). These results suggest that the direction of the adversarial image perturbation, in combination with a specific image, is perceptually relevant to features that the human visual system uses to classify objects. These findings thus give evidence that strong black box adversarial attacks can transfer from CNNs to humans, and show remarkable similarities between failure cases of CNNs and human vision.

In all cases, the average response time was longer for the `adv` condition relative to the other conditions (Figure Supp.2b), though this result was only statistically significant for two comparisons. If this trend remains predictive, it would seem to contradict the case when we presented `false` images (Figure Supp.2a). One interpretation is that in the `false` case, the transfer of adversarial features to humans accompanied by more confidence, whereas here the transfer was accompanied by less confidence, possibly due to competing adversarial and true class features in the `adv` condition.

## 5 Discussion

Our results invite several questions that we discuss briefly.

### 5.1 Have we actually fooled human observers or did we change the true class?

One might naturally wonder whether we have fooled the human observer or whether we have replaced the input image with an image that actually belongs to a different class. In our work, the perturbations we made were small enough that they generally do not change the output class for a human who has no time limit (the reader may verify this by observing Figures 1a,b, 2c, and Supp.5 through Supp.8). We can thus be confident that we did not change the true class of the image, and that we really did fool the time-limited human. Future work aimed at fooling humans with no time-limit will need to tackle the difficult problem of obtaining a better ground truth signal than visual labeling by humans.

### 5.2 How do the adversarial examples work?

We did not design controlled experiments to prove that the adversarial examples work in any specific way, but we informally observed a few apparent patterns illustrated in Figure 3: disrupting object edges, especially by mid-frequency modulations perpendicular to the edge; enhancing edges both by increasing contrast and creating texture boundaries; modifying texture; and taking advantage of dark regions in the image, where the perceptual magnitude of small $\epsilon$ perturbations can be larger.

### 5.3 What are the implications for machine learning security and society?

The fact that our transfer-based adversarial examples fool time-limited humans but not no-limit humans suggests that the lateral and top-down connections used by the no-limit human are relevant to human robustness to adversarial examples. This suggests that machine learning security research should explore the significance of these top-down or lateral connections further. One possible explanation for our observation is that no-limit humans are fundamentally more robust to adversarial example and achieve this robustness via top-down or lateral connections. If this is the case, it could point the way to the development of more robust machine learning models. Another possible explanation is that no-limit humans remain highly vulnerable to adversarial examples but adversarial examples do not transfer from feed-forward networks to no-limit humans because of these architectural differences.

Our results suggest that there is a risk that imagery could be manipulated to cause human observers to have unusual reactions; for example, perhaps a photo of a politician could be manipulated in a way that causes it to be perceived as unusually untrustworthy or unusually trustworthy in order to affect the outcome of an election.

### 5.4 Future Directions

In this study, we designed a procedure that according to our hypothesis would transfer adversarial examples to humans. An interesting set of questions relates to how sensitive that transfer is to different elements of our experimental design. For example: How does transfer depend on $\epsilon$? Was model ensembling crucial to transfer? Can the retinal preprocessing layer be removed? We suspect that retinal preprocessing and ensembling are both important for transfer to humans, but that $\epsilon$ could be made smaller. See Figure Supp.4 for a preliminary exploration of these questions.

## 6 Conclusion

Susceptibility to adversarial examples has been widely assumed – in the absence of experimental evidence – to be a property of machine learning classifiers, but not of human judgement. In this work, we correct this assumption by showing that adversarial examples based on perceptible but class-preserving perturbations that fool multiple machine learning models also fool time-limited humans. Our findings demonstrate striking similarities between the decision boundaries of convolutional neural networks and the human visual system. We expect this observation to lead to advances in both neuroscience and machine learning research.

## Acknowledgements

We are grateful to Ari Morcos, Bruno Olshausen, David Sussillo, Hanlin Tang, John Cunningham, Santani Teng, and Daniel Yamins for useful discussions. We also thank Dan Abolafia Simon Kornblith, Katherine Lee, Kathryn Rough, Niru Maheswaranathan, Catherine Olsson, David Sussillo, and Santani Teng, for helpful feedback on the manuscript. We thank Google Brain residents for useful feedback on the work. We also thank Deanna Chen, Leslie Philips, Sally Jesmonth, Phing Turner, Melissa Strader, Lily Peng, and Ricardo Prada for assistance with IRB and experiment setup.

## Footnotes

[2]The study was granted an Institutional Review Board (IRB) exemption by an external, independent, ethics board (Quorum review ID 33016).

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
