[Supplementary Material]

# Supplemental material

## A   Supplementary Figures and Tables

Figure Supp.1: **Adversarial examples optimized on more models / viewpoints sometimes appear more meaningful to humans.** This observation is a clue that machine-to-human transfer may be possible. (a) A canonical example of an adversarial image reproduced from [13]. This adversarial attack has moderate but limited ability to fool the model after geometric transformations or to fool models other than the model used to generate the image. (b) An adversarial attack causing a cat image to be labeled as a computer while being robust to geometric transformations, adopted from [1]. Unlike the attack in a, the image contains features that seem semantically computer-like to humans. (c) An adversarial patch that causes images to be labeled as a toaster, optimized to cause misclassification from multiple viewpoints, reproduced from [4]. Similar to b, the patch contains features that appear toaster-like to a human.

Figure Supp.2: **Reaction time.** (a) subjects average response time to `false` images. (b) response time of subjects for the `adv`, `image`, and `flip` conditions (error bars $\pm$ SE; * reflects $p < 0.05$; two sample two-tailed t-test).

Figure Supp.3: **Stronger adversarial effects corresponds to faster reaction time, possibly indicating increased confidence in decision.** Plots show probability of choosing the adversarially targeted class, when the true image class is not one of the choices that subjects can report, estimated by averaging the responses of all subjects (two-tailed t-test relative to chance level $0.5$; error bars $\pm$ SE; *: $p < 0.05$; **: $p < 0.01$; ***: $p < 0.001$). The probability of choosing the targeted label is computed by binning trials within percentile reaction time ranges (0-33 percentile, 33-67 percentile, and 67-100 percentile). The bias relative to chance level of $0.5$ is significant when people reported their decision quickly (i.e., when they are more confident), but not significant when they reported their decision more slowly.

Figure Supp.4: **Intuition on factors contributing to the transfer to humans.** To give some intuition on the factors contributing to transfer, we examine a cat image from ImageNet (*(a)* left) that is already perceptually close to the target adversarial dog class, making the impact of subtle adversarial effects more obvious even on long observation (*(a)* right). Note that this image was not used in the experiment, and that typical images in the experiment *did not* fool unconstrained humans. *(b)* shows adversarial images with different perturbation sizes ranging from $\epsilon = 8$ to $\epsilon = 40$. Even smaller perturbation $\epsilon = 8$ make the adversarial image perceptually more similar to a dog image, which suggests that transfer to humans may be robust to small $\epsilon$. *(c)* Investigation of the importance of matching initial visual processing. The adversarial image on the left is similar to a dog, while removing the retina layer leads to an image which is less similar to a dog. This suggests that matching initial processing is an important factor in transferring adversarial examples to humans. *(d)* Investigation of the importance of the number of models in the ensemble. We generated adversarial images with $\epsilon = 32$ using an ensemble of size $1-10$ models. One can see that adversarial perturbations become markedly less similar to a dog class as the number of models in the ensemble is reduced. This supports the importance of ensembling to the transfer of adversarial examples to humans.

Table Supp.1: **Adversarial examples transfer to humans.** Number of subjects that reported the correct class of images in the `adv` condition with lower mean accuracy compared to their mean accuracy in the `image` and `flip` conditions.

| Group | adv $<$ image | adv $<$ flip | total |
|---|---|---|---|
| pets | 29 | 22 | 35 |
| hazard | 19 | 16 | 24 |
| vegetables | 21 | 23 | 32 |

Table Supp.2: **Accuracy of models on ImageNet validation set.** $^*$ models trained on ImageNet with retina layer pre-pended and with train data augmented with rescaled images in the range of $[40, 255 - 40]$; $^{**}$ model trained with adversarial examples augmented data. First ten models are models used in the adversarial training ensemble. Last two models are models used to test the transferability of adversarial examples.

| Model | Top-1 accuracy |
|---|---|
| Resnet V2 101 | 0.77 |
| Resnet V2 101$^*$ | 0.7205 |
| Inception V4 | 0.802 |
| Inception V4$^*$ | 0.7518 |
| Inception Resnet V2 | 0.804 |
| Inception Resnet V2$^*$ | 0.7662 |
| Inception V3 | 0.78 |
| Inception V3$^*$ | 0.7448 |
| Resnet V2 152 | 0.778 |
| Resnet V2 50$^*$ | 0.708 |
| Resnet V2 50 (test) | 0.756 |
| Inception V3$^{**}$ (test) | 0.776 |

Table Supp.3: **Accuracy of ensemble used to generate adversarial examples on images at different conditions.** $^*$ models trained on ImageNet with retina layer appended and with train data augmented with rescaled images in the range of $[40, 255 - 40]$; Numbers triplet reflects accuracy on images from pets, hazard, and vegetables groups, respectively.

| Train Model | adv (%) | flip (%) |
|---|---|---|
| Resnet V2 101 | 0.0, 0.0, 0.0 | 95, 92, 91 |
| Resnet V2 101$^*$ | 0.0, 0.0, 0.0 | 87, 87, 77 |
| Inception V4 | 0.0, 0.0, 0.0 | 96, 95, 86 |
| Inception V4$^*$ | 0.0, 0.0, 0.0 | 87, 87, 73 |
| Inception Resnet V2 | 0.0, 0.0, 0.0 | 97, 95, 95 |
| Inception Resnet V2$^*$ | 0.0, 0.0, 0.0 | 87, 83, 73 |
| Inception V3 | 0.0, 0.0, 0.0 | 97, 94, 89 |
| Inception V3$^*$ | 0.0, 0.0, 0.0 | 83, 86, 74 |
| Resnet V2 152 | 0.0, 0.0, 0.0 | 96, 95, 91 |
| Resnet V2 50$^*$ | 0.0, 0.0, 0.0 | 82, 85, 81 |

Table Supp.4: **Accuracy of test models on images at different conditions.** $^{**}$ model trained on both clean and adversarial images. Numbers triplet is accuracy on pets, hazard, and vegetables groups, respectively.

| Model | adv (%) | flip (%) |
|---|---|---|
| Resnet V2 50 | 8.7, 9.4, 13 | 93, 91, 85 |
| Inception V3$^{**}$ | 6.0, 6.9, 17 | 95, 92, 94 |

Table Supp.5: **Attack success on model ensemble.** Same convention as Table Supp.3

| Model | adv (%) | flip (%) |
|---|---|---|
| Resnet V2 101 | 100, 100, 100 | 2, 0, 0 |
| Resnet V2 101* | 100, 100, 100 | 3, 0, 0 |
| Inception V4 | 100, 100, 100 | 1, 0, 1 |
| Inception V4* | 100, 100, 100 | 4, 1, 0 |
| Inception Resnet V2 | 100, 100, 100 | 1, 0, 1 |
| Inception Resnet V2* | 100, 100, 100 | 5, 2, 0 |
| Inception V3 | 100, 100, 100 | 1, 0, 0 |
| Inception V3* | 100, 100, 100 | 5, 1, 1 |
| Resnet V2 152 | 100, 100, 100 | 1, 0, 0 |
| Resnet V2 50* | 100, 100, 100 | 3, 1, 0 |

Table Supp.6: **Attack success on test models.** ** model trained on both clean and adversarial images. Numbers triplet is error on pets, hazard, and vegetables groups, respectively.

| Model | adv (%) | flip (%) |
|---|---|---|
| Resnet V2 50 | 87, 85, 57 | 1.3, 0.0, 0.0 |
| Inception V3** | 89, 87, 74 | 1.5, 0.5, 0.0 |

## B  Details of retinal blurring layer

### B.1  Computing the primate eccentricity map

Let $d_{viewer}$ be the distance (in meters) of the viewer from the display and $d_{hw}$ be the height and width of a square image (in meters). For every spatial position (in meters) $c = (x, y) \in R^2$ in the image we compute the retinal eccentricity (in radians) as follows:

$$\theta(c) = \tan^{-1}(\frac{||c||_2}{d_{viewer}}) \tag{1}$$

and turn this into a target resolution in units of radians

$$r_{rad}(c) = \min(\alpha\theta(c), \beta). \tag{2}$$

We then turn this target resolution into a target spatial resolution in the plane of the screen,

$$r_m(c) = r_{rad}(c)\left(1 + \tan^2(\theta(c))\right), \tag{3}$$

$$r_{pixel}(c) = r_m(c) \cdot [\text{pixels per meter}]. \tag{4}$$

This spatial resolution for two point discrimination is then converted into a corresponding low-pass cutoff frequency, in units of cycles per pixel,

$$f(c) = \frac{\pi}{r_{pixel}}, \tag{5}$$

where the numerator is $\pi$ rather than $2\pi$ since the two point discrimination distance $r_{pixel}$ is half the wavelength.

Finally, this target low-pass spatial frequency $f(c)$ for each pixel is used to linearly interpolate each pixel value from the corresponding pixel in a set of low pass filtered images, as described in the following algorithm (all operations on matrices are assumed to be performed elementwise), We additionally cropped $X_{retinal}$ to 90% width before use, to remove artifacts from the image edge.

Note that because the per-pixel blurring is performed using linear interpolation into images that were low-pass filtered in Fourier space, this transformation is both fast to compute and fully differentiable.

## C  Calculating probability of coarse class

To calculate the probability a model assigns to a coarse class, we summed probabilities assigned to the individual classes within the coarse class. Let $S_{\text{target}}$ be the set of all individual labels

---

**Algorithm 1** Applying retinal blur to an image

---

1: $X_{img} \leftarrow$ input image
2: $F \leftarrow$ image containing corresponding target lowpass frequency for each pixel, computed from $f(c)$
3: $\tilde{X} \leftarrow$ FFT($X_{img}$)
4: $G \leftarrow$ norm of spatial frequency at each position in $Y$
5: CUTOFF_FREQS $\leftarrow$ list of frequencies to use as cutoffs for low-pass filtering
6: **for** $f'$ in CUTOFF_FREQS **do**
7: $\quad \tilde{Y}_{f'} \leftarrow \tilde{X} \odot \exp\left(-\frac{G^2}{f^2}\right)$
8: $\quad Y_f \leftarrow$ InverseFFT($\tilde{Y}_{f'}$)
9: **end for**
10: $w(c) \leftarrow$ linear interpolation coefficients for $F(c)$ into CUTOFF_FREQS $\quad \forall c$
11: $X_{retinal}(c) \leftarrow \sum_{f'} w_{f'}(c) Y_{f'}(c) \quad \forall c$

---

in the target coarse class. Let $S_{\text{other}}$ be all other individual labels not in the target coarse class. $|S_{\text{target}}| + |S_{\text{other}}| = 1000$, since there are 1000 labels in ImageNet. Let $Y$ be the coarse class variable and $y_{\text{target}}$ be our target coarse class. We can compute the probability a model $k$ assigns to a coarse class given image $X$ as

$$P_k(Y = y_{\text{target}}|X) = \sum_{i \in S_{\text{target}}} P_k(Y = y_i|X) = \sigma\left(\log \frac{\sum_{i \in S_{\text{target}}} \tilde{F}_k(i|X)}{\sum_{i \in S_{\text{other}}} \tilde{F}_k(i|X)}\right) \qquad (6)$$

where $\tilde{F}_k(i|X)$ is the unnormalized probability assigned to fine class $i$ (in practice = $\exp(logits)$ of class $i$). The coarse logit of the model with respect to the target class $y_{\text{target}}$ is then $F_k(Y = y_{\text{target}}|X) = \log \frac{\sum_{i \in S_{\text{target}}} \tilde{F}_k(i|X)}{\sum_{i \in S_{\text{other}}} \tilde{F}_k(i|X)}$.

## D  Adversarial images generation.

In the pipeline, an image is drawn from the source coarse class and perturbed to be classified as an image from the target coarse class. The attack method we use, the iterative targeted attack [24], is performed as

$$\tilde{X}_{adv}^n = X_{adv}^{n-1} - \alpha * \text{sign}(\nabla_{X^n}(J(X^n|y_{\text{target}}))),$$
$$X_{adv}^n = \text{clip}\left(\tilde{X}_{adv}^n, [X - \epsilon, X + \epsilon]\right), \qquad (7)$$

where $J$ is the cost function as described below, $y_{\text{target}}$ is the label of the target class, $\alpha$ is the step size, $X_{adv}^0 = X$ is the original clean image, and $X_{adv} = X_{adv}^N$ is the final adversarial image. We set $\alpha = 2$, and $\epsilon$ is given per-condition in Section 3.2.2. After optimization, any perturbation whose $\ell_\infty$-norm was less than $\epsilon$ was scaled to have $\ell_\infty$-norm of $\epsilon$, for consistency across all perturbations.

Our goal was to create adversarial examples that transferred across many ML models before assessing their transferability to humans. To accomplish this, we created an ensemble from the geometric mean of several image classifiers, and performed the iterative attack on the ensemble loss [26]

$$J(X|y_{target}) = -\log[P_{\text{ens}}(y_{\text{target}}|X)], \qquad (8)$$
$$P_{\text{ens}}(y|X) \propto \exp(\mathbb{E}_k[\log P_k(y|X)]), \qquad (9)$$

where $P_k(y|X)$ is the coarse class probabilities from model $k$, and $P_{\text{ens}}(y|X)$ is the probability from the ensemble. In practice, $J(X|y_{target})$ is equivalent to standard cross entropy loss based on coarse logits averaged across models in the ensemble (see Appendix C for the coarse logit definition).

To encourage a high transfer rate, we retained only adversarial examples that were successful against all 10 models for the `adv` condition and at least 7/10 models for the `false` condition (see Section 3.2.2 for condition definitions).

# E   Convolutional Neural Network Models

Some of the models in our ensemble are from a publicly available pretrained checkpoints[3], and others are our own instances of the architectures, specifically trained for this experiment on ImageNet with the retinal layer prepended. To encourage invariance to image intensity scaling, we augmented each training batch with another batch with the same images but rescaled in the range of $[40, 255 - 40]$, instead of $[0, 255]$. Supplementary Table Supp.2 identifies all ten models used in the ensemble, and shows their top-1 accuracies, along with two holdout models that we used for evaluation.

Figure Supp.5: **Adversarial Examples for false condition** (a) pets group. (b) hazard group. (c) vegetables group.

Figure Supp.6: **Adversarial Examples** pets group

Figure Supp.7: **Adversarial Examples** hazard group

Figure Supp.8: **Adversarial Examples** vegetables group

## F  Image List from Imagenet

The specific imagenet images used from each class in the experiments in this paper are as follows:

**dog:**

'n02106382_564.JPEG',          'n02110958_598.JPEG',          'n02101556_13462.JPEG',
'n02113624_7358.JPEG',         'n02113799_2538.JPEG',         'n02091635_11576.JPEG',
'n02106382_2781.JPEG',          'n02112706_105.JPEG',          'n02095570_10951.JPEG',

'n02093859_5274.JPEG',      'n02109525_10825.JPEG',      'n02096294_1400.JPEG',
'n02086646_241.JPEG',       'n02098286_5642.JPEG',       'n02106382_9015.JPEG',
'n02090379_9754.JPEG',      'n02102318_10390.JPEG',      'n02086646_4202.JPEG',
'n02086910_5053.JPEG',      'n02113978_3051.JPEG',       'n02093859_3809.JPEG',
'n02105251_2485.JPEG',      'n02109525_35418.JPEG',      'n02108915_7834.JPEG',
'n02113624_430.JPEG',       'n02093256_7467.JPEG',       'n02087046_2701.JPEG',
'n02090379_8849.JPEG',      'n02093754_717.JPEG',        'n02086079_15905.JPEG',
'n02102480_4466.JPEG',      'n02107683_5333.JPEG',       'n02102318_8228.JPEG',
'n02099712_867.JPEG',       'n02094258_1958.JPEG',       'n02109047_25075.JPEG',
'n02113624_4304.JPEG',      'n02097474_10985.JPEG',      'n02091032_3832.JPEG',
'n02085620_859.JPEG',       'n02110806_582.JPEG',        'n02085782_8327.JPEG',
'n02094258_5318.JPEG',      'n02087046_5721.JPEG',       'n02095570_746.JPEG',
'n02099601_3771.JPEG',      'n02102480_41.JPEG',         'n02086910_1048.JPEG',
'n02094114_7299.JPEG',      'n02108551_13160.JPEG',      'n02110185_9847.JPEG',
'n02097298_13025.JPEG',     'n02097298_16751.JPEG',      'n02091467_555.JPEG',
'n02113799_2504.JPEG',      'n02085782_14116.JPEG',      'n02097474_13885.JPEG',
'n02105251_8108.JPEG',      'n02113799_3415.JPEG',       'n02095570_8170.JPEG',
'n02088238_1543.JPEG',      'n02097047_6.JPEG',          'n02104029_5268.JPEG',
'n02100583_11473.JPEG',     'n02113978_6888.JPEG',       'n02104365_1737.JPEG',
'n02096177_4779.JPEG',      'n02107683_5303.JPEG',       'n02108915_11155.JPEG',
'n02086910_1872.JPEG',      'n02106550_8383.JPEG',       'n02088094_2191.JPEG',
'n02085620_11897.JPEG',     'n02096051_4802.JPEG',       'n02100735_3641.JPEG',
'n02091032_1389.JPEG',      'n02106382_4671.JPEG',       'n02097298_9059.JPEG',
'n02107312_280.JPEG',       'n02111889_86.JPEG',         'n02113978_5397.JPEG',
'n02097209_3461.JPEG',      'n02089867_1115.JPEG',       'n02097658_4987.JPEG',
'n02094114_4125.JPEG',      'n02100583_130.JPEG',        'n02112137_5859.JPEG',
'n02113799_19636.JPEG',     'n02088094_5488.JPEG',       'n02089078_393.JPEG',
'n02098413_1794.JPEG',      'n02113799_1970.JPEG',       'n02091032_3655.JPEG',
'n02105855_11127.JPEG',     'n02096294_3025.JPEG',       'n02094114_4831.JPEG',
'n02111889_10472.JPEG',     'n02113624_9125.JPEG',       'n02097474_9719.JPEG',
'n02094433_2451.JPEG',      'n02095889_6464.JPEG',       'n02093256_458.JPEG',
'n02091134_2732.JPEG',      'n02091244_2622.JPEG',       'n02094114_2169.JPEG',
'n02090622_2337.JPEG',      'n02101556_6764.JPEG',       'n02096051_1459.JPEG',
'n02087046_9056.JPEG',      'n02098105_8405.JPEG',       'n02112137_5696.JPEG',
'n02110806_7949.JPEG',      'n02097298_2420.JPEG',       'n02085620_6814.JPEG',
'n02108915_1703.JPEG',      'n02100877_19273.JPEG',      'n02106550_3765.JPEG',
'n02107312_3524.JPEG',      'n02111889_2963.JPEG',       'n02113624_9129.JPEG',
'n02097047_3200.JPEG',      'n02093256_8365.JPEG',       'n02093991_9420.JPEG',
'n02112137_1635.JPEG',      'n02111129_3530.JPEG',       'n02101006_8123.JPEG',
'n02102040_5033.JPEG',      'n02113624_437.JPEG',        'n02090622_5866.JPEG',
'n02110806_3711.JPEG',      'n02112137_14788.JPEG',      'n02105162_7406.JPEG',
'n02097047_5061.JPEG',      'n02108422_11587.JPEG',      'n02091467_4265.JPEG',
'n02091467_12683.JPEG',     'n02104365_3628.JPEG',       'n02086646_3314.JPEG',
'n02099849_736.JPEG',       'n02100735_8112.JPEG',       'n02112018_12764.JPEG',
'n02093428_11175.JPEG',     'n02110627_9822.JPEG',       'n02107142_24318.JPEG',
'n02105162_5489.JPEG',      'n02093754_5904.JPEG',       'n02110958_215.JPEG',
'n02095314_4027.JPEG',      'n02109961_3250.JPEG',       'n02108551_7343.JPEG',
'n02110627_10272.JPEG',     'n02088364_3099.JPEG',       'n02110806_2721.JPEG',
'n02095314_2261.JPEG',      'n02106550_9870.JPEG',       'n02107574_3991.JPEG',
'n02095570_3288.JPEG',      'n02086079_39042.JPEG',      'n02096294_9416.JPEG',
'n02110806_6528.JPEG',      'n02088466_11397.JPEG',      'n02092002_996.JPEG',
'n02098413_8605.JPEG',      'n02085620_712.JPEG',        'n02100236_3011.JPEG',
'n02086646_7788.JPEG',      'n02085620_4661.JPEG',       'n02098105_1746.JPEG',
'n02113624_8608.JPEG',      'n02097474_1168.JPEG',       'n02107683_1496.JPEG',
'n02110185_12849.JPEG',     'n02085620_11946.JPEG',      'n02087394_16385.JPEG',
'n02110806_22671.JPEG',     'n02113624_526.JPEG',        'n02096294_12642.JPEG',
'n02113023_7510.JPEG',      'n02088364_13285.JPEG',      'n02095889_2977.JPEG',
'n02105056_9215.JPEG',      'n02102318_9744.JPEG',       'n02097298_11834.JPEG',

'n02111277_16201.JPEG', 'n02085782_8518.JPEG', 'n02113978_11280.JPEG',
'n02106382_10700.JPEG'.

**cat:**

'n02123394_661.JPEG', 'n02123045_11954.JPEG', 'n02123394_3695.JPEG',
'n02123394_2692.JPEG', 'n02123597_12166.JPEG', 'n02123045_7014.JPEG',
'n02123159_2777.JPEG', 'n02123394_684.JPEG', 'n02124075_543.JPEG',
'n02123597_7557.JPEG', 'n02124075_7857.JPEG', 'n02123597_3770.JPEG',
'n02124075_4986.JPEG', 'n02123045_568.JPEG', 'n02123394_1541.JPEG',
'n02123597_3498.JPEG', 'n02123597_10304.JPEG', 'n02123394_2084.JPEG',
'n02123597_5283.JPEG', 'n02123597_13807.JPEG', 'n02124075_12282.JPEG',
'n02123597_8575.JPEG', 'n02123045_11787.JPEG', 'n02123394_888.JPEG',
'n02123045_1815.JPEG', 'n02123394_7614.JPEG', 'n02123597_27865.JPEG',
'n02124075_1279.JPEG', 'n02123394_4775.JPEG', 'n02123394_976.JPEG',
'n02123394_8385.JPEG', 'n02123597_14791.JPEG', 'n02123045_10424.JPEG',
'n02123597_7698.JPEG', 'n02124075_8140.JPEG', 'n02123045_3754.JPEG',
'n02123597_1819.JPEG', 'n02123597_395.JPEG', 'n02123394_415.JPEG',
'n02124075_9747.JPEG', 'n02123045_9467.JPEG', 'n02123159_6842.JPEG',
'n02123394_9611.JPEG', 'n02123597_7283.JPEG', 'n02123597_11799.JPEG',
'n02123597_660.JPEG', 'n02123045_7511.JPEG', 'n02123597_10723.JPEG',
'n02123159_7836.JPEG', 'n02123597_14530.JPEG', 'n02123597_28555.JPEG',
'n02123394_6079.JPEG', 'n02123394_6792.JPEG', 'n02123597_11564.JPEG',
'n02123597_8916.JPEG', 'n02124075_123.JPEG', 'n02123045_5150.JPEG',
'n02124075_353.JPEG', 'n02123597_12941.JPEG', 'n02123045_10095.JPEG',
'n02123597_6533.JPEG', 'n02123045_4611.JPEG', 'n02123597_754.JPEG',
'n02123394_8561.JPEG', 'n02123597_6409.JPEG', 'n02123159_4909.JPEG',
'n02123597_564.JPEG', 'n02123394_1633.JPEG', 'n02123394_1196.JPEG',
'n02123394_2787.JPEG', 'n02124075_10542.JPEG', 'n02123597_6242.JPEG',
'n02123597_3063.JPEG', 'n02123597_13164.JPEG', 'n02123045_7449.JPEG',
'n02123045_13299.JPEG', 'n02123394_8165.JPEG', 'n02123394_1852.JPEG',
'n02123597_8771.JPEG', 'n02123159_6581.JPEG', 'n02123394_5906.JPEG',
'n02124075_2747.JPEG', 'n02124075_11383.JPEG', 'n02123597_3919.JPEG',
'n02123394_2514.JPEG', 'n02124075_7423.JPEG', 'n02123394_6968.JPEG',
'n02123045_4850.JPEG', 'n02123045_10689.JPEG', 'n02124075_13539.JPEG',
'n02123597_13378.JPEG', 'n02123159_4847.JPEG', 'n02123394_1798.JPEG',
'n02123597_27951.JPEG', 'n02123159_587.JPEG', 'n02123597_1825.JPEG',
'n02123159_2200.JPEG', 'n02123597_12.JPEG', 'n02123597_6778.JPEG',
'n02123597_6693.JPEG', 'n02123045_11782.JPEG', 'n02123597_13706.JPEG',
'n02123394_9032.JPEG', 'n02124075_4459.JPEG', 'n02123597_13752.JPEG',
'n02123394_2285.JPEG', 'n02123597_1410.JPEG', 'n02123159_6134.JPEG',
'n02123597_11290.JPEG', 'n02123597_6347.JPEG', 'n02123394_1789.JPEG',
'n02123045_11255.JPEG', 'n02123394_6096.JPEG', 'n02123394_4081.JPEG',
'n02123394_5679.JPEG', 'n02123394_2471.JPEG', 'n02123159_5797.JPEG',
'n02123597_13894.JPEG', 'n02124075_10854.JPEG', 'n02123394_8605.JPEG',
'n02124075_8281.JPEG', 'n02123597_11724.JPEG', 'n02123394_8242.JPEG',
'n02123394_3569.JPEG', 'n02123597_10639.JPEG', 'n02123045_3818.JPEG',
'n02124075_6459.JPEG', 'n02123394_185.JPEG', 'n02123597_8961.JPEG',
'n02124075_9743.JPEG', 'n02123394_1627.JPEG', 'n02123597_13175.JPEG',
'n02123045_2694.JPEG', 'n02123597_4537.JPEG', 'n02123597_6400.JPEG',
'n02123045_7423.JPEG', 'n02123597_3004.JPEG', 'n02123394_2988.JPEG',
'n02124075_9512.JPEG', 'n02123394_6318.JPEG', 'n02123597_1843.JPEG',
'n02124075_2053.JPEG', 'n02123597_3828.JPEG', 'n02123394_14.JPEG',
'n02123394_8141.JPEG', 'n02124075_1624.JPEG', 'n02123597_459.JPEG',
'n02124075_6405.JPEG', 'n02123045_8595.JPEG', 'n02123159_3226.JPEG',
'n02124075_9141.JPEG', 'n02123597_2031.JPEG', 'n02123045_2354.JPEG',
'n02123597_6710.JPEG', 'n02123597_6613.JPEG', 'n02123159_1895.JPEG',
'n02123394_2953.JPEG', 'n02123394_5846.JPEG', 'n02123394_513.JPEG',
'n02123045_16637.JPEG', 'n02123394_7848.JPEG', 'n02123394_3229.JPEG',
'n02123045_8881.JPEG', 'n02123394_8250.JPEG', 'n02124075_7651.JPEG',

'n02123394_200.JPEG', 'n02123394_2814.JPEG', 'n02123045_6445.JPEG',
'n02123394_2467.JPEG', 'n02123045_3317.JPEG', 'n02123597_1422.JPEG',
'n02123597_13442.JPEG', 'n02123394_8225.JPEG', 'n02123597_9337.JPEG',
'n02123394_32.JPEG', 'n02123394_2193.JPEG', 'n02123394_1625.JPEG',
'n02123597_8799.JPEG', 'n02123597_13241.JPEG', 'n02123597_7681.JPEG',
'n02123597_4550.JPEG', 'n02123597_3896.JPEG', 'n02123394_9554.JPEG',
'n02124075_13600.JPEG', 'n02123394_571.JPEG', 'n02123597_10886.JPEG',
'n02123045_6741.JPEG', 'n02123045_10438.JPEG', 'n02123045_9954.JPEG'.

**spider:**

'n01775062_517.JPEG', 'n01774750_18017.JPEG', 'n01774384_13186.JPEG',
'n01774750_3115.JPEG', 'n01775062_5075.JPEG', 'n01773549_1541.JPEG',
'n01775062_4867.JPEG', 'n01775062_8156.JPEG', 'n01774750_7128.JPEG',
'n01775062_4632.JPEG', 'n01773549_8734.JPEG', 'n01773549_2274.JPEG',
'n01773549_10298.JPEG', 'n01774384_1811.JPEG', 'n01774750_7498.JPEG',
'n01774750_10265.JPEG', 'n01773549_1964.JPEG', 'n01774750_3268.JPEG',
'n01773549_6095.JPEG', 'n01775062_8812.JPEG', 'n01774750_10919.JPEG',
'n01775062_1180.JPEG', 'n01773549_7275.JPEG', 'n01773549_9346.JPEG',
'n01773549_8243.JPEG', 'n01775062_3127.JPEG', 'n01773549_10608.JPEG',
'n01773549_3442.JPEG', 'n01773157_1487.JPEG', 'n01774750_7775.JPEG',
'n1775062_419.JPEG', 'n01774750_7638.JPEG', 'n01775062_847.JPEG',
'n01774750_3154.JPEG', 'n01773549_1534.JPEG', 'n01773157_1039.JPEG',
'n01775062_5644.JPEG', 'n01775062_8525.JPEG', 'n01773797_216.JPEG',
'n01775062_900.JPEG', 'n01774750_8513.JPEG', 'n01774750_3424.JPEG',
'n01774750_3085.JPEG', 'n01775062_3662.JPEG', 'n01774384_15681.JPEG',
'n01774750_326.JPEG', 'n01773157_9503.JPEG', 'n01774750_3332.JPEG',
'n01774750_2799.JPEG', 'n01773157_10606.JPEG', 'n01773157_1905.JPEG',
'n01773549_379.JPEG', 'n01773797_597.JPEG', 'n01773157_3226.JPEG',
'n01774750_7875.JPEG', 'n01774384_16102.JPEG', 'n01773549_2832.JPEG',
'n01775062_5072.JPEG', 'n01773549_4278.JPEG', 'n01773549_5854.JPEG',
'n01774384_1998.JPEG', 'n01774750_13875.JPEG', 'n01775062_8270.JPEG',
'n01773549_2941.JPEG', 'n01774750_5235.JPEG', 'n01773549_4150.JPEG',
'n01774750_6217.JPEG', 'n01775062_3137.JPEG', 'n01774750_5480.JPEG',
'n01774384_11955.JPEG', 'n01775062_8376.JPEG', 'n01773157_2688.JPEG',
'n01773549_6825.JPEG', 'n01774750_10422.JPEG', 'n01774384_20786.JPEG',
'n01773549_398.JPEG', 'n01773549_4965.JPEG', 'n01774750_7470.JPEG',
'n01775062_1379.JPEG', 'n01774384_2399.JPEG', 'n01773549_9799.JPEG',
'n01775062_305.JPEG', 'n01774384_15519.JPEG', 'n01774750_3333.JPEG',
'n01774750_2604.JPEG', 'n01774750_3134.JPEG', 'n01774750_4646.JPEG',
'n01775062_5009.JPEG', 'n01774750_10200.JPEG', 'n01775062_7964.JPEG',
'n01774384_2458.JPEG', 'n01773797_3333.JPEG', 'n01774750_9987.JPEG',
'n01773549_5790.JPEG', 'n01773549_854.JPEG', 'n01774750_11370.JPEG',
'n01774750_10698.JPEG', 'n01774750_9287.JPEG', 'n01773797_6703.JPEG',
'n01773797_931.JPEG', 'n01773549_5280.JPEG', 'n01773797_5385.JPEG',
'n01773797_1098.JPEG', 'n01774750_436.JPEG', 'n01774384_13770.JPEG',
'n01774750_9780.JPEG', 'n01774750_8640.JPEG', 'n01774750_653.JPEG',
'n01774384_12554.JPEG', 'n01774750_9716.JPEG'

**snake:**

'n01737021_7081.JPEG', 'n01728572_16119.JPEG', 'n01735189_10620.JPEG',
'n01751748_3573.JPEG', 'n01729322_6690.JPEG', 'n01735189_20703.JPEG',
'n01734418_4792.JPEG', 'n01749939_2784.JPEG', 'n01729977_4113.JPEG',
'n01756291_6505.JPEG', 'n01742172_3003.JPEG', 'n01728572_19317.JPEG',
'n01739381_5838.JPEG', 'n01737021_1381.JPEG', 'n01749939_4704.JPEG',
'n01755581_10792.JPEG', 'n01729977_9474.JPEG', 'n01744401_11909.JPEG',
'n01739381_10303.JPEG', 'n01749939_820.JPEG', 'n01728572_27743.JPEG',
'n01734418_12057.JPEG', 'n01742172_8636.JPEG', 'n01729977_14112.JPEG',
'n01739381_6286.JPEG', 'n01734418_761.JPEG', 'n01740131_13437.JPEG',
'n01728920_9571.JPEG', 'n01753488_4234.JPEG', 'n01749939_5712.JPEG',

'n01739381_6072.JPEG', 'n01739381_7683.JPEG', 'n01729322_9202.JPEG',
'n01751748_13413.JPEG', 'n01756291_4626.JPEG', 'n01742172_9733.JPEG',
'n01737021_12610.JPEG', 'n01739381_87.JPEG', 'n01729977_1134.JPEG',
'n01753488_637.JPEG', 'n01748264_18478.JPEG', 'n01728572_22360.JPEG',
'n01737021_3386.JPEG', 'n01751748_560.JPEG', 'n01751748_18223.JPEG',
'n01749939_5750.JPEG', 'n01748264_7044.JPEG', 'n01739381_1163.JPEG',
'n01751748_311.JPEG', 'n01756291_9028.JPEG', 'n01739381_10473.JPEG',
'n01728572_1415.JPEG', 'n01729322_10918.JPEG', 'n01748264_653.JPEG',
'n01753488_10957.JPEG', 'n01756291_3990.JPEG', 'n01756291_11915.JPEG',
'n01756291_6776.JPEG', 'n01740131_11661.JPEG', 'n01729977_5715.JPEG',
'n01737021_16733.JPEG', 'n01753488_15197.JPEG', 'n01744401_7248.JPEG',
'n01728572_7661.JPEG', 'n01740131_13680.JPEG', 'n01729322_5446.JPEG',
'n01749939_6508.JPEG', 'n01748264_2140.JPEG', 'n01729977_16782.JPEG',
'n01748264_7602.JPEG', 'n01756291_17857.JPEG', 'n01729977_461.JPEG',
'n01742172_20552.JPEG', 'n01735189_3258.JPEG', 'n01728920_9265.JPEG',
'n01748264_18133.JPEG', 'n01748264_16699.JPEG', 'n01739381_1006.JPEG',
'n01753488_10555.JPEG', 'n01751748_3202.JPEG', 'n01734418_3929.JPEG',
'n01751748_5908.JPEG', 'n01751748_8470.JPEG', 'n01739381_3598.JPEG',
'n01739381_255.JPEG', 'n01729977_15657.JPEG', 'n01748264_21477.JPEG',
'n01751748_2912.JPEG', 'n01728920_9154.JPEG', 'n01728572_17552.JPEG',
'n01740131_14560.JPEG', 'n01729322_5947.JPEG'.

**Broccoli:**

'n07714990_8640.JPEG', 'n07714990_5643.JPEG', 'n07714990_7777.JPEG',
'n07714990_888.JPEG', 'n07714990_3398.JPEG', 'n07714990_4576.JPEG',
'n07714990_8554.JPEG', 'n07714990_1957.JPEG', 'n07714990_4201.JPEG',
'n07714990_3130.JPEG', 'n07714990_4115.JPEG', 'n07714990_524.JPEG',
'n07714990_6504.JPEG', 'n07714990_3125.JPEG', 'n07714990_5838.JPEG',
'n07714990_1779.JPEG', 'n07714990_6393.JPEG', 'n07714990_1409.JPEG',
'n07714990_4962.JPEG', 'n07714990_7282.JPEG', 'n07714990_7314.JPEG',
'n07714990_11933.JPEG', 'n07714990_1202.JPEG', 'n07714990_3626.JPEG',
'n07714990_7873.JPEG', 'n07714990_3325.JPEG', 'n07714990_3635.JPEG',
'n07714990_12524.JPEG', 'n07714990_14952.JPEG', 'n07714990_7048.JPEG',
'n07714990_500.JPEG', 'n07714990_7950.JPEG', 'n07714990_2445.JPEG',
'n07714990_1294.JPEG', 'n07714990_7336.JPEG', 'n07714990_14743.JPEG',
'n07714990_1423.JPEG', 'n07714990_2185.JPEG', 'n07714990_6566.JPEG',
'n07714990_567.JPEG', 'n07714990_1532.JPEG', 'n07714990_5212.JPEG',
'n07714990_8971.JPEG', 'n07714990_6116.JPEG', 'n07714990_5462.JPEG',
'n07714990_7644.JPEG', 'n07714990_8596.JPEG', 'n07714990_1138.JPEG',
'n07714990_15078.JPEG', 'n07714990_1602.JPEG', 'n07714990_2460.JPEG',
'n07714990_159.JPEG', 'n07714990_9445.JPEG', 'n07714990_471.JPEG',
'n07714990_1777.JPEG', 'n07714990_9760.JPEG', 'n07714990_1528.JPEG',
'n07714990_12338.JPEG', 'n07714990_2201.JPEG', 'n07714990_6850.JPEG',
'n07714990_4492.JPEG', 'n07714990_7791.JPEG', 'n07714990_9752.JPEG',
'n07714990_1702.JPEG', 'n07714990_3682.JPEG', 'n07714990_14342.JPEG',
'n07714990_2661.JPEG', 'n07714990_5467.JPEG'.

**Cabbage:**

'n07714571_14784.JPEG', 'n07714571_4795.JPEG', 'n07714571_11969.JPEG',
'n07714571_1394.JPEG', 'n07714571_4155.JPEG', 'n07714571_3624.JPEG',
'n07714571_13753.JPEG', 'n07714571_7351.JPEG', 'n07714571_10316.JPEG',
'n07714571_7235.JPEG', 'n07714571_17716.JPEG', 'n07714571_1639.JPEG',
'n07714571_5107.JPEG', 'n07714571_4109.JPEG', 'n07714571_11878.JPEG',
'n07714571_15910.JPEG', 'n07714571_14401.JPEG', 'n07714571_2741.JPEG',
'n07714571_8576.JPEG', 'n07714571_1624.JPEG', 'n07714571_13479.JPEG',
'n07714571_2715.JPEG', 'n07714571_3676.JPEG', 'n07714571_12371.JPEG',
'n07714571_4829.JPEG', 'n07714571_3922.JPEG', 'n07714571_10377.JPEG',
'n07714571_8040.JPEG', 'n07714571_8147.JPEG', 'n07714571_10377.JPEG',
'n07714571_8040.JPEG', 'n07714571_5730.JPEG', 'n07714571_16460.JPEG',

'n07714571_8198.JPEG', 'n07714571_1095.JPEG', 'n07714571_3922.JPEG', 'n07714571_7745.JPEG', 'n07714571_6301.JPEG'.

## Footnotes

[3]https://github.com/tensorflow/models/tree/master/research/slim