[Reviews · NeurIPS 2018]

Reviewer 1



This paper introduces an interesting experiment that tries to show that adversarial examples that transfer across recently proposed deep learning models can influence the visual classification produced by time-limited humans (at most 2.2 to 2.5 seconds for looking at the image and making the classification). The idea is to take a set of 38 observers to classify images into 2 classes from within each of three separate groups (pets - cat or dog, hazard - snake or spider, vegetables - broccoli or cabbage). The images are presented in different ways: 1) original image, 2) adversarial image manipulated to be classified by an ensemble classification method (of 10 recently proposed deep learning models), 3) flip, which is the original image manipulated by a vertically flipped adversarial perturbation (this is a control case to show that low signal-to-noise ratio alone does not explain poor human classification), and 4) false, which is an image outside the three groups above, manipulated by an adversarial perturbation to be classified as one of the classes within one of the three groups. In general, results show that 1) adversarial perturbations in the false class successfully biased human classification towards the target class; and 2) adversarial perturbations cause observers to select the incorrect class even when the correct class is available (this is the 2nd manipulation described above). I found this paper quite interesting to read and I support its publication. I have a couple of comments that may influence the final writing of the paper. Can there be an error due to mistakes in pressing the buttons given that the assignment of classes to buttons was randomized? Was there some calibration phase before each experiment to rule that issue out? Another experiment would be to measure how humans perform as time limit increases or decreases?

Reviewer 2



This paper shows that adversarial examples that transfer across computer vision models can successfully influence the perception of human observers. This is not a typical machine learning paper, and the main claim is verified empirically through well designed human psychophysics experiments. The paper is well written and easy to follow. The experiment configuration and reasoning are solid, and the conclusion from the paper is significant and might lead to advance the design of new algorithms and NN architectures to defend adversarial attacks. Since this is not a typical machine learning paper and more of experimental psychophysics (which is out of my domain), my confidence score isn't very high (3). The implications of this work could be significant.

Reviewer 3



This paper demonstrate that adversarial examples can be transferred from computer vision model to time-limited human observers. Although the explorations are interesting, the results are based on the strong assumption of time-limited human. This makes the contribution somewhat limited, and also makes some of the findings questionable. For example, in the experimental setup, the target class “dog” resembles the visual appearance of “cat” more, compared to those false examples with randomly chosen classes. The observation of significant correlation between target class and user reported class could be the effect that dog images and cat images look alike to human vision system if shown for very short period of time. It’s unclear to me whether in all conditions, the source-target pairs are constructed to be visually similar, which caused the human perception bias. I also found the findings less practically useful, given that human generally perceive the world without time constraint. But I do like the implications of the study for better improving machine learning security by exploring lateral connections.